# Assessment of Lipopeptide Mixtures Produced by *Bacillus subtilis* as Biocontrol Products against Apple Scab (*Venturia inaequalis*)

**DOI:** 10.3390/microorganisms10091810

**Published:** 2022-09-09

**Authors:** Aline Leconte, Ludovic Tournant, Jérôme Muchembled, Jonathan Paucellier, Arnaud Héquet, Barbara Deracinois, Caroline Deweer, François Krier, Magali Deleu, Sandrine Oste, Philippe Jacques, François Coutte

**Affiliations:** 1UMRt BioEcoAgro 1158-INRAE, Equipe Métabolites Secondaires D’origine Microbienne, Institut Charles Viollette, Université de Lille, F-59000 Lille, France; 2JUNIA, UMRt BioEcoAgro 1158-INRAE, Equipe Métabolites Spécialisés D’origine Végétale, Institut Charles Viollette, F-59000 Lille, France; 3UMRt BioEcoAgro 1158-INRAE, Equipe Métabolites Secondaires D’origine Microbienne, TERRA Teaching and Research Centre, Gembloux Agro-Bio Tech, Université de Liège, B-5030 Gembloux, Belgium; 4FREDON Hauts-de-France-265 rue Becquerel, F-62750 Loos-en-Gohelle, France; 5Lipofabrik SAS, 917 rue des Saules, F-59343 Lesquin, France

**Keywords:** *Bacillus subtilis*, lipopeptides, surfactin, mycosubtilin, fengycin, *Venturia inaequalis*, apple scab, biopesticide

## Abstract

Apple scab is an important disease conventionally controlled by chemical fungicides, which should be replaced by more environmentally friendly alternatives. One of these alternatives could be the use of lipopeptides produced by *Bacillus subtilis*. The objective of this work is to study the action of the three families of lipopeptides and different mixtures of them in vitro and in vivo against *Venturia inaequalis*. Firstly, the antifungal activity of mycosubtilin/surfactin and fengycin/surfactin mixtures was determined in vitro by measuring the median inhibitory concentration. Then, the best lipopeptide mixture ratio was produced using Design of Experiment (DoE) to optimize the composition of the culture medium. Finally, the lipopeptides mixtures efficiency against *V. inaequalis* was assessed in orchards as well as the evaluation of the persistence of lipopeptides on apple. In vitro tests show that the use of fengycin or mycosubtilin alone is as effective as a mixture, with the 50–50% fengycin/surfactin mixture being the most effective. Optimization of culture medium for the production of fengycin/surfactin mixture shows that the best composition is glycerol coupled with glutamic acid. Finally, lipopeptides showed in vivo antifungal efficiency against *V. inaequalis* regardless of the mixture used with a 70% reduction in the incidence of scab for both mixtures (fengycin/surfactin or mycosubtilin/surfactin). The reproducibility of the results over the two trial campaigns was significantly better with the mycosubtilin/surfactin mixture. The use of *B. subtilis* lipopeptides to control this disease is very promising.

## 1. Introduction

Apple scab is the most important disease affecting apple orchards worldwide, both in terms of control and commercial losses [1,2,3,4]. The pathogen, *Venturia inaequalis*, is a hemibiotrophic ascomycete fungus with an asexual and a sexual cycle. The best time to infect apple trees is in the spring. Ejected ascospores germinate to form mycelium. This mycelium will multiply infections in the summer by asexual cycle. Then the sexual cycle starts again in autumn to protect itself from winter. 

Fruits affected by the disease will not be marketable and affected trees will be weakened [3,5]. Despite the use of prophylactic methods (varietal choice), the application of fungicides remains predominant. Apple scab can be controlled by the use of fungicides such as multisite fungicides (captan, copper), or single-site fungicides (succinate dehydrogenase inhibitor, demethylation inhibitors). However, more and more cases of resistance or reduced sensitivity to single-site fungicides have been reported worldwide [5,6,7,8,9,10,11]. This massive use of chemical fungicides has serious environmental consequences (pollution, residues, resistance). Moreover, in organic agriculture, apple scab can be controlled using copper or sulphur. Nevertheless, the excessive application of copper is responsible for important environmental concerns [12]. It is therefore essential to develop alternative control methods, such as biopesticides. 

Several studies have investigated the use of biocontrol products to overcome resistance and pollution problems. Among these promising biocontrol products, *Bacillus*’ lipopeptides represent very good alternative to chemical pesticides. *Bacillus subtilis* is a Gram-positive bacterium from soil. The manipulation of its genome or specific feeding strategies makes it possible to act on the regulation and orientation of its metabolism towards the production of molecules of interest [13,14,15,16,17,18]. *B. subtilis* can produce lipopeptides by a non-ribosomal mechanism, which have interesting biological properties. Various studies have already shown the antimicrobial activities of these molecules [17,19,20,21,22,23]. There is a great diversity of these molecules, with more than 100 different structures [24]. These are classified into three distinct families according to their amino acid part: Fengycins (fengycins and plipastatins), Iturins (mycosubtilins, iturins and bacillomycins) and Surfactins (surfactins, pumilacidins, lichenysins). Surfactins are cyclic lipopeptides composed of 7 amino acid residues and a β-hydroxy fatty acid residue differing from each other in their peptide moieties and their fatty acid length. Surfactins are mainly known for their biosurfactant activity. Some studies have suggested an antifungal activity for surfactins against different fungi [23,25,26]. However, they are mostly known for a synergistic effect when combined with fengycin or mycosubtilin [23,27,28]. Moreover, studies have shown the existence of antiviral, antitumour, anticoagulant activities and stimulation of plant defense mechanisms [19,21,29,30,31]. Fengycins are composed of 10 amino acids (partially cyclic peptide) and of a β-hydroxylated lipid chain. According to the diversity of the peptide part, different subgroups have been identified: fengycin A and B, plipastatin A and B, and agrastatin A and B. These molecules have antifungal activities and are able to inhibit the growth of a large number of plant pathogens, especially filamentous fungi [23,32,33,34,35,36,37]. Iturins family consists of a group of cyclic lipopeptides composed of a heptapeptide moiety and a chain of fatty acids linked by an α-amino bond. These are known for their strong antifungal activity against pathogenic yeasts and fungi [22,32,38].

In addition to their individual activities, synergistic antifungal activities have also been reported for lipopeptide mixtures [23,28]. The production and use of lipopeptide mixtures therefore have several interests. Firstly, it allows the use of non-genetically modified strains, and secondly, it has been shown that certain lipopeptide families can act synergistically against phytopathogens. Synergistic effects have been demonstrated in vitro [32,39], but also in vivo against *Zymoseptoria tritici*, *Botrytis cinerea* or *Bremia lactucae* [28,40,41]. Production of lipopeptide families by *B. subtilis* is widely described in the scientific literature [24]. This production is strain-dependent and can be influenced by the growth conditions (pH, temperature, oxygen transfer and composition of culture medium). Fickers et al. (2008) have, for example, shown the impact of temperature on the production of mycosubtilin isoforms by the natural strain ATCC 6633 [42]. Studies on the production of surfactin and fengycin by the *B. subtilis* strain ATCC 21332, and its derivatives BBG21, had clearly shown that a high oxygen transfer favours the production of surfactin over fengycin and vice versa [43,44]. Carbon and nitrogen sources also have a significant impact on lipopeptide production. 

For example, it was shown that the use of mannitol as a carbon source or a mixture of urea and ammonium sulphate as a nitrogen source had a significant effect on the specific production of fengycin in *B. subtilis* ATCC 21332 derivatives [45]. Urea was also use as nitrogen source to optimize the production of lipopeptides by *B. subtilis* SPB1 [46]. Other authors have shown the impact of glycerol and arginine on lipopeptide production in wild type *B. amyloliquefaciens* 0G [47]. Optimization of the culture medium for lipopeptide production through the use of experimental design (DoE) is a very effective method. The Plackett–Burman design was used to optimize lipopeptide production by *B. subtilis* S499 [48]. Response Surface Methodology (RSM) was carried out to optimize the production of iturin by *B. subtilis* BH072 [16] or the production of surfactin by *B. subtilis* BBG131 [49]. Tagushi experimental design was performed to enhance surfactin production by *B. subtilis* ATCC 21332 [50].

The main objective of this work is to study the interest of using lipopeptide mixture (i.e., fengycin/surfactin and mycosubtilin/surfactin) produced by different natural strains of *B. subtilis* to manage apple scab in Organic Agriculture. To achieve this objective, we first determined the in vitro efficacy of the three families of lipopeptide alone and of the ratios in these two different mixtures against *V. inaequalis*. Secondly, the culture medium was optimized in order to produce a fengycin/surfactin mixture (FS) with an effective ratio against apple scab. This production was then tested in orchards during 2 trial campaigns in comparison with another mixture of lipopeptides containing mycosubtilin/surfactin (MS) produced at a semi-industrial scale, a copper/sulphur treatment and a commercial preparation of *B. subtilis*’s spores. Finally, the persistence of the lipopeptides on the fruit was investigated. This scientific strategy is summarized in a schematic representation in Figure 1.

## 2. Materials and Methods

### 2.1. Lipopeptide Production and Purification for In Vitro Antifungal Assays

The three families of lipopeptides (surfactin, fengycin and mycosubtilin) used to performed the in vitro experiments against *V. inaequalis* are presented in Table 1, they were produced and purified using genetically modified mono-producers strains of *B. subtilis* as recently described [51]. These experiments are presented in the following paragraph.

### 2.2. In Vitro Experiments against V. inaequalis

The S755 strain of *V. inaequalis* is used to perform microplate assays [23,52]. A spore suspension in glucose peptone (1.43% glucose and 0.71% bactopeptone) is obtained after 20 days of culture under malt agar medium in the dark. Lipopeptide activity is assessed by a 96-well microplate assay in liquid medium. Powdered lipopeptides are solubilized by adjusting for 100% purity in dimethyl sulfoxide (DMSO 100%) at a concentration of 60 g/L. After solubilization, two types of mixtures, fengycin–surfactin (FS) and mycosubtilin–surfactin (MS) are made with various mass proportions 100%, 80–20%, 60–40% and 50–50% (Table 1). A range of 15 concentrations in glucose peptone culture medium is performed according to the modalities in the Table 1. Each concentration is distributed in six wells per line with 140 µL per well. The spore suspension of *V. inaequalis* is distributed in four wells per line with 60 µL per well. The first two wells have only 60 µL of glucose peptone culture medium and are used as controls. The microplate is sealed and shaken at 140 rpm for 6 days at 20 °C in the dark. After six days of incubation, OD values are obtained by a microplate reader at 365 nm. A non-linear regression is used to determine the median inhibitory concentration (IC_50_).

### 2.3. Medium Optimization for the Production of a Mix Fengycin/Surfactin by B. subtilis ATCC 21332

In order to produce the best FS mixture, *B. subtilis* ATCC 21332 strain was used. The objective of this work was to determine the best composition of the culture medium to produce a FS ratio close to 50–50% (as determined in in vitro experiments against *V. inaequalis*). DoE were used to optimize the culture medium on a base of Landy’s medium as described previously [51].

According to the information found in the literature and our experience in lipopeptide production, we design an optimized full factorial plan involving four variables: the carbon source; the nitrogen source; the phosphate concentration; and the volumetric oxygen transfer coefficient. The first two variables are divided into three levels: three sources of carbon (glucose 40 g/L, mannitol 40 g/L and glycerol 40 g/L) and three sources of nitrogen (glutamic acid 5 g/L, arginine 1.48 g/L and a urea–ammonium mixture 1.6 g/L). The next two variables are in only two levels: two KH_2_PO_4_ concentrations (1 g/L and 2 g/L) and two different volumetric oxygen transfer coefficient (K_L_a) values (86 h^−1^ and 135 h^−1^). These variables were investigated to determine the most influencing parameters for the production of fengycin, surfactin, biomass, specific fengycin production (Y_Pf/X_), specific surfactin production (Y_Ps/X_) and the best percentage of fengycin produced. The interactions between the variables were also investigated. The software Minitab^®^ 18 (Minitab LLC, State College, PA, USA) was used to design this factorial plan and to analyze the results. This plan makes it possible to obtain 16 different media with 2 volumetric oxygen transfer coefficient conditions each, i.e., 32 conditions in triplicate, i.e., 96 samples instead of 108, for a complete non-optimized factorial plan.

All these experiments were conducted in triplicate using a Biolector^®^ microfermentation system (m2p-labs GmbH, Baesweiler, Germany) in 48-well microtiter Flowerplates incubated during 72 h at 30 °C, 800 rpm and pH 7.0 buffered with 0.1 M MOPS [53]. In order to obtain the two different K_L_a, the wells are filled with either 800 µL (k_L_a = 135 h^−1^ or 1.5 mL (k_L_a = 86 h^−1^) of culture medium [54]. Biomass, pH and dissolved oxygen were monitored on-line. Before the culture in Biolector^®^ a preculture procedure was done. Briefly, a first preculture is made from a glycerol stock suspension on LB agar medium at 30 °C. Then, one colony of this culture on a Petri dish is subcultured to inoculate 5 mL of LB medium, incubated at 30 °C and shaken at 250 rpm for 24 h. This first liquid preculture is then used to inoculate a second preculture of 50 mL of Landy medium at pH 7 buffered with 0.1M MOPS in an Erlenmeyer flask and incubated at 30 °C under 160 rpm of agitation. This second pre-culture is stopped in the exponential growth phase (OD_600nm_ < 4) and used to inoculate the microtiter Flowerplates. After Biolector culture, culture broths were collected, the pH of each well was checked and adjusted to 7.0 with KOH if necessary, then centrifuged at 10,000× *g* for 10 min. Supernatant were then analyzed by RP-HPLC, as described below.

### 2.4. Quantification of Lipopeptides Using RP-UPLC

Before analysis, the supernatant is diluted by half with ethanol and centrifuged at 10,000× *g* for 10 min. Analysis was performed by RP-UPLC according to the recently described protocol [51]. Briefly, 10 µL of each sample is then analyzed using ACQUITY UPLC system (Waters, Milford, MA, USA) equipped with C18 column (UP5TP18-250/030 C18, Interchim, Montluçon, France) and coupled to a UV detector (detection at 214 nm). The mobile phase consisted of an acetonitrile/water/TFA gradient and at a flow rate of 0.6 mL/min. Lipopeptides were quantified using standards of surfactin and fengycin supplied by Sigma Aldrich (Sigma-Aldrich, Saint-Louis, MO, USA) and standard of mycosubtilin supplied by Lipofabrik (Lipofabrik, Lesquin, France).

### 2.5. Lipopeptide Production for Orchard Trials

The different lipopeptide preparations tested in these trials were produced by two different *B. subtilis* strains. The natural strain *B. subtilis* ATCC 21332 was used to produce the FS mixture. The strain was grown in 5 L Erlenmeyer flasks in previously optimized medium at 30 °C, with 130 rpm agitation and a *v*/*v* ratio of 20% medium in Erlenmeyer flask, which results in a K_L_a around 100 h^−1^, according to the equation proposed by Fahim et al. (2012) [43]. The composition of the culture medium used was as follows: glycerol, 40 g/L; glutamic acid, 5 g/L; yeast extract, 1 g/L; KH_2_PO_4_ 1 g/L; MgSO_4_, 0.5 g/L; KCl, 0.5 g/L; CuSO_4_, 1.6 mg/L; Fe_2_(SO_4_)_3_, 1.2 mg/L; MnSO_4_, 0.4 mg/L. The medium was buffered with 0.1 M MOPS at pH 7.0. After 72 h of culture, the lipopeptides were purified by a sequential process as developed by Coutte et al. [44] and more recently described in detail [51]. Briefly, the cells are firstly removed by a centrifugation step. The supernatant is then concentrated 10 times by an ultrafiltration step on a 10 kDa regenerated cellulose Hydrosart ultrafiltration membrane (Sartorius, Goettingen, Germany). Four water diafiltration steps follow to purify the lipopeptides retained by the membrane. Finally, a final ultrafiltration step in the presence of 70% (*v*/*v*) ethanol is performed, which allows to break the lipopeptide micelles and to pass them into the permeate. The ethanolic permeate containing the lipopeptides is then concentrated by evaporation of ethanol. A mixture containing FS in solution was obtained. Concentration of this mixture was then adjusted at 250 mg/L of total lipopeptide in DMSO 0.1% before its use in orchards.

Another natural strain of *B. subtilis* selected by the company Lipofabrik was used to produce the MS mixture in a confidential industrial process. A mixture containing MS in solution (80–20%) was obtained. Concentration of this mixture was then adjusted at 500 mg/L of total lipopeptides in DMSO 0.1% before its use in orchards. The choice of this test concentration was guided by the supplier’s recommendations.

In order to be applied for the protection of apple orchards, the lipopeptides treatments solutions are prepared as follows. Lipopeptide mixture solutions containing either fengycin (55%) and surfactin (45%) or mycosubtilin (80%) and surfactin (20%) were supplemented with adjuvant. Adjuvants were added extemporaneously before treatment at a concentration of 0.2% each. Heliosol^®^ supplied by Actionpin (Actionpin, Castets, France) was used as adjuvant for the FS mixture and a combination of AEG and CMC supplied from Lamberti (Lamberti SPA, Gallarate, Italy) was used as adjuvant for the MS mixture.

### 2.6. Protection of Apple Trees against Scab

Lipopeptide mixtures were then evaluated in orchards to protect apple trees against natural infestation of *V. inaequalis*. The trials conducted in 2018 and 2019 are a grouping of seven modalities in a Fisher block design with four replicates and a water control. The elementary plots were composed of three trees of the Jonagold variety (several mutants evenly distributed) and a ‘buffer’ tree of the Jugala variety, in order to limit possible spray drift. Both varieties are highly susceptible to the disease. The trial was set up in the experimental orchard of FREDON Hauts-de-France on the site of Loos-en-Gohelle (F-62). Under natural conditions, this orchard represented a situation favorable to the disease, with significant rates of contamination by the disease in previous “classic” years. All the treatments were carried out as a preventive measure at a maximum rate of 7 days. Protection was renewed as soon as the threshold of 20 mm of rainfall was crossed.

Among the modalities tested in 2018, one plant protection product (SERENADE, *B. subtilis* str. QST 713) guided the evaluation programme towards an evaluation of the different substances during the period of primary contamination in the post-flowering situation, in order to respect the maximum number of applications. Protection against scab was achieved by means of copper and/or sulphur applications during the pre-flowering and flowering phase. For all the modalities (except for water control), a same pre-flowering treatment was applied as described in the Table 2.

The seven modalities studied (Table 2) were a water control, the adjuvant control of the mixture of fengycin and surfactin, the adjuvant control of the mixture of mycosubtilin and surfactin, an organic farming reference (copper and/or sulphur), the SERENADE specialty (Bayer Crop Science), the modality of the mixture of mycosubtilin and surfactin, the modality of the mixture of fengycin and surfactin.

In 2019, the modalities were repeated, this time with protection applied over the entire period of primary infections using the lipopeptide mixtures. The seven modalities studied (Table 3) were a water control, the adjuvant control of the mixture of fengycin and surfactin, the adjuvant control of the mixture of mycosubtilin and surfactin, an organic farming reference (copper and/or sulphur), a modality based on fertilizer specialties (containing, in particular, copper), the modality composed of mycosubtilin and surfactin, and the modality composed of fengycin and surfactin.

### 2.7. Detection of Lipopeptide on Apple Using QuEChERS Method and Study of the Persistence of Lipopeptides

The analysis of lipopeptide residues was conducted on apples taken after the first trial campaign (2018) in the orchards described above. Twenty apples were harvested 45 days after the last treatment and were stored for 15 days at 4 °C before analysis. Control samples (untreated), samples treated with the MS mixture, samples treated with the FS mixture were analyzed. Among these 20 apples, 5 apples were randomly selected and the residues were extracted according to the QuEChERS method regulated by NF EN 15662 May 2018 [55]. Two types of matrix were distinguished, either the skin after peeling and grinding, or the apple flesh after grinding. This method is a versatile method for the determination of pesticide residues from plant material by GC-MS and LC-MS/MS with extraction/partitioning with acetonitrile. In our case, we replaced the cleaning step by adding PSA (Primary-Secondary Amine) with a step of Solid Phase Extraction (SPE) on a 1 g cartridge of C18 according to the diagram presented in Appendix A. After a concentration step using centrifuge evaporator (miVac, Gene Vac, Ipswich, UK) during 2 h at 40 °C, samples were analyzed by RP-UPLC-MS according to the protocol described below. 

### 2.8. Quantification of the Lipopeptides by RP-UPLC and Mass Spectrometry (RP-UPLC-MS)

Fengycin, surfactin and mycosubtilin working standards were weighed out and dissolved in methanol/H_2_O/formic acid (50/49.9/0.1) to prepare different solutions at concentration ranging between 0.02 µg/µL to 0.33 µg/µL for each lipopeptide in order to quantify lipopeptides ranging between 0.004 to 0.0666 mg of lipopeptides by kg of apple. Dried samples were dissolved in 200 µL of methanol/H_2_O/formic acid (50/49.9/0.1) and centrifuged for 10 min at 8000− *g*. A volume of 10 µL of sample or calibrators were chromatographically separated at 30 °C on an ACQUITY UPLC BEH C18 column (130 Å, 1.7 um, 2.1 × 50 mm, Waters Corporation) with the following acetonitrile gradient at 0.5 mL/min (the mobile phases consisted of solvent A (0.1% (*v*/*v*) formic acid/99.9% (*v*/*v*) water) and solvent B (0.1% (*v*/*v*) formic acid/99.9% (*v*/*v*) acetonitrile): from 35% to 50% solvent B over 10 min, from 50% to 90% solvent B over 5 min followed by washing and equilibrating procedures with respectively 95% and 35% solvent B during 5 min. The eluate was directed into the electrospray ionization source of the ACQUITY QDa mass spectrometer (Waters Corporation). The cone voltage and the capillary voltage were set to 15 V and 0.8 kV, respectively. MS data were collected for *m*/*z* values in the range of 30 and 1 250 Da with a sampling rate of 10 points/sec.

The RP-UPLC-MS data were analyzed with Empower 3 software (Waters Corporation). Quantification was performed with the *m*/*z* 753.44, 1072.69 and 1107.58. The ion of *m*/*z* 753.44, corresponding to the most intense ion detected for fengycin in the samples, is associated to the diprotonated isoform of fengycin A C19 or B C17. The ion of *m*/*z* 1072.69, corresponding to the most intense ion detected for surfactin in the samples, is associated to the sodium adduct of a C16 isoform. The ion of *m*/*z* 1107.58, corresponding to the most intense ion detected for mycosubtilin in the samples, is associated to the sodium adduct of a C17 isoform. Under this chromatographic condition, the retention times for these three ions were, respectively, 9.7, 16.3 and 5.0 min. Extracted chromatograms from these three ions were generated and integrated. To determine the concentration of the three lipopeptides in the different samples, the areas under the mass peak were taken to establish a linear or polynomial relation with concentration in standard (coefficient of determination (R^2^) of 0.9969, 09887 and 0.9993, respectively).

### 2.9. Statistical Analysis

In the in vitro test on the efficiency of lipopeptides against *V. inaequalis*, six different wells of each modality were performed, the average of these six values is presented in Figure 2 and Figure 3 as well as the confidence interval for each value.

The statistical analysis of the experimental design detailed in part 2.3 of the Materials and Methods section was carried out using Minitab^®^ 18 software (Minitab LLC, State College, PA, USA). A two-sided 95% confidence interval was defined. Analysis of variance (ANOVA) and a linear factorial regression analysis were performed to evaluate the significant of each factor in the different output parameters (i.e., for the production of fengycin, surfactin, biomass, specific fengycin production (Y_Pf/X_), specific surfactin production (Y_Ps/X_) and the best percentage of fengycin produced). The quality of the linear regression model was analyzed using the Fisher F test and the coefficient of determination (R^2^). Pareto plots, main effects and interactions are obtained on the standardized effects through analysis in Minitab^®^ 18 software.

During orchard efficiency testing, several observations were made in orchards during the cycle of primary contamination of the disease. The last one took place after the release of the last stains from the primary contaminations. The notations were carried out on the three central trees of each elementary plot in the four repetitions of the test. In each elementary plot, 210 leaves were observed to count the number of leaves attacked. A leaf was considered attacked as soon as a stain appeared (all-or-nothing notation). This choice is justified by the fact that at this stage of the evaluation of lipopeptides, the primary objective was to observe the effectiveness against the disease and not the development of a control strategy with the assessment of the attack’s severity. In total, the sampling effort focused on 840 leaves per modality distributed at the rate of seven shoots of 10 leaves per tree, distributed over the different leaf stages of the tree (top–middle–bottom).

From these observations, statistical analysis of variables and interpretation of results were done. Analyses of the effectiveness of the tested active substances were based on the percentage of leaves attacked by the disease. The sequence of analysis began with verification of the realistic aspect of the trial, namely, its ability to provide useful data and the achievement of consistent results. After verifying a sufficient level of disease development in the water control modality and obtaining consistent results for the substances against the water control modality, the data were subjected to an analysis of variance. This analysis was followed by a test of Newman and Keuls to compare the preparations with each other (at the threshold α = 5%).

Comparisons of the different modalities tested were made regarding the reference modality of the test and the water control modality, to specify the level of contamination of the culture during the observations. To do this, the efficiency was calculated based on Abbott’s efficiency calculation method: Efficiency = 100 × T_0_ − T_t_/T_0_ where T_0_ = percentage of attack in the reference plot T_t_ = percentage of attack in the plot studied.

## 3. Results

The apple scab biocontrol trials in this work were carried out on an organic farming plot requiring the production of lipopeptides by natural non-genetically modified *B. subtilis* strains. In most cases, natural *B. subtilis* strains produce mixtures of several lipopeptide families (surfactin and fengycin or surfactin and iturin, or all three families together [24]). Previous work on the pathogen *V. inaequalis* has shown the good efficiency of fengycin (alone or in a mixture with surfactin) and of mycosubtilin (alone or in a mixture with surfactin) during in vitro experiments [23]. The choice therefore fell on the production of a mixture of FS and a mixture of MS.

As presented in Figure 1, the efficiency of the lipopeptide mixture ratio was first measured in vitro. In a second step, we optimized the production of the best FS ratio by the strain *B. subtilis* ATCC 21332 using a full factorial design plan. Then, this FS mixture of lipopeptides was applied in organic orchard to fight against apple scab and compared with MS mixture and conventional reference products. Finally, the remanence of lipopeptides on harvested and stored fruits was evaluated.

### 3.1. Antifungal Activities of the Mixture of Lipopeptides against V. inaequalis

The IC_50_ for the mixture FS at different proportions (*w*/*w*) is presented Figure 2. It can be observed that fengycin alone (F100) with an IC_50_ of 0.05 mg/L (0.03–0.07) presents a better antifungal activity against *V. inaequalis* S755 strain than surfactin alone (S100) with an IC_50_ of 6.38 mg/L (5.30–7.67). The mixtures F80–S20 (0.04 mg/L (0.03–0.05)), F60–S40 (0.05 mg/L (0.04–0.06)), and F50–S50 (0.08 mg/L (0.07–0.09)) have IC_50_ similar to fengycin alone. When the proportion of surfactin is higher than fengycin (F40–S60 0.23 mg/L (0.18–0.29) and F20–S80 0.17 mg/L (0.15–0.20)), the IC_50_ increases but remains lower than the IC_50_ of surfactin alone. From these results, the FS mixture 50–50% was selected for future experiments.

Figure 3 presents the IC_50_ for the MS mixture at different proportions. With an IC_50_ at 1.38 mg/L (1.15–1.68), mycosubtilin alone (M100) has a better activity than surfactin alone (S100) (4.79 mg/L (3.81–6.01)).

If one compares the surfactin results obtained in these two independent experiments (Figure 2 and Figure 3), the results show that the IC_50_ of surfactin are slightly different, nevertheless the confidence intervals overlap. The M80–S20 mixture is the most efficient mixture with an IC_50_ of 0.83 mg/L (0.83–1.12). An increase in median inhibitory concentration can be seen for the last two concentrations containing the highest amount of surfactin. However, the mixtures M60–S40 (1.23 mg/L (1.16–1.29)), M50–S50 (1.73 mg/L (1.26–2.36)) and M40–S60 (1.42 mg/L (1.18–1.72)) have similar IC_50_ to each other and to mycosubtilin alone. With 2.70 mg/L (2.64–2.76), the M20–S80 mixture has an activity between M40–S60 and surfactin alone. From these results the MS mixture 80–20% was selected for future experiments.

**Figure 2 microorganisms-10-01810-f002:**
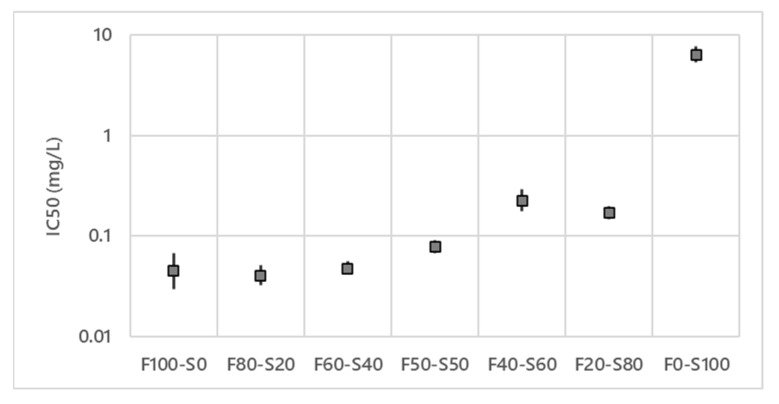
Effect of the lipopeptide ratios (*w*/*w*) on the IC_50_ of fengycin/surfactin mixtures on *V. inaequalis* S755 strain. IC_50_ in logarithmic scale.

**Figure 3 microorganisms-10-01810-f003:**
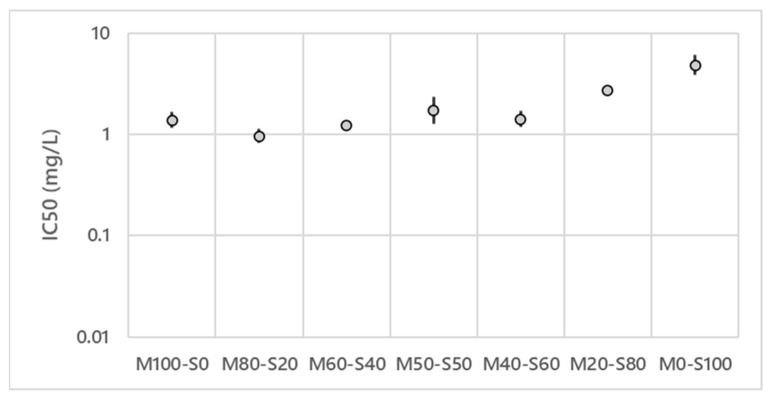
Effect of the lipopeptide ratios (*w*/*w*) on the IC_50_ of mycosubtilin/surfactin mixtures on *V. inaequalis* S755 strain. Logarithmic scale.

### 3.2. Medium Optimization Using DoE for the Production of Fengycin/surfactin Mixture by B. subtilis ATCC 21332

An optimized full factorial design was performed to determine the factors influencing the production of surfactin or fengycin in the *B. subtilis* ATCC 21332 strain and particularly the percentage of fengycin produced in relation to total lipopeptides (i.e., fengycin + surfactin). The different factors investigated were the carbon source (glucose, mannitol, glycerol), the nitrogen source (glutamic acid, arginine, urea + ammonium sulfate), the phosphate concentration (1 or 2 g/L) and the K_L_a (86 h^−1^ or 135 h^−1^). Design and analysis of the results were carried out using the Minitab^®^ tool18. The results are presented in Figure 4 and Figure 5.

In Figure 4, it can be observed that the three main factors (carbon or nitrogen sources and phosphate concentration) as well as the interaction between carbon and nitrogen sources have a significant impact on biomass production. Combinations between glucose/arginine or mannitol/urea + ammonium present the best biomass production over 5 g of DW/L (data not shown). The effect identified as significant on the final fengycin concentration, and its specific production is the interaction between the carbon source and nitrogen source factors (Figure 4). From these results, it can be also observed that the final fengycin concentration and the specific production of this lipopeptide are not necessarily significantly impacted by the same factors. Carbon source and phosphate concentration alone have a significant impact on both, which is not the case of nitrogen source. Nitrogen source alone has a significant impact only on the specific production of fengycin, as well as K_L_a. On the other hand, the fengycin concentration is also influenced by the interaction between the carbon source or nitrogen source and the phosphate concentration. However, these effects are less significant than the carbon x nitrogen interaction.

In order to further investigate the interactions and their effects on the fengycin specific production, interaction diagrams were made (Figure 5). The interaction between two factors is studied level by level. For the interaction between the two factors carbon source and nitrogen source, it turns out that the best option for specific production is the combination of glycerol as carbon source with glutamic acid as nitrogen source. For this same medium, the use of phosphate at a concentration of 1 g/L gives the best results. On the other hand, another combination, namely, glycerol, arginine and phosphate at 1 g/L, gives a result in specific production less important but close to that observed for this last medium.

These results show that mannitol has a positive effect on the specific production of fengycin compared to the reference composition of the Landy’s medium (i.e., glucose, glutamic acid, phosphate at 1 g/L). It allows the latter to reach 21.5 mg/g compared to 19.3 mg/g for the control (glucose). Nevertheless, glycerol was found to have the best impact on fengycin specific production (26.9 mg/g). Glutamic acid and arginine present the best results on the fengycin specific production. With regard to the percentage of fengycin produced, the four factors have a significant importance with the carbon source and the nitrogen source having the same strong impact. Interestingly, the effect of K_L_a is particularly important on this parameter, suggesting a good optimization lever. The interactions between the carbon and nitrogen sources, as well as between the nitrogen source and the phosphate concentration are also marked. Once again, the results presented in Figure 5 show that the best medium to increase the percentage of fengycin should contain glycerol as a carbon source (nearly 50% of fengycin produced).

However, the combination of glycerol with glutamic acid or arginine shows similar results. The phosphate concentration in these cases does not show much difference. Direct impacts of the different factors are presented in Appendix A.

The study of the main effects impacting the final surfactin concentration and its specific production shows that three of them are common: carbon source, nitrogen source and the interaction of these two factors (Figure 4). On the other hand, two other effects, unique to each response, are significant: K_L_a and phosphate concentration on the final concentration and surfactin specific production, respectively. The study of the interaction levels for specific surfactin production (Figure 5) reveals that the best medium composition corresponds to a combination of glucose with the urea + ammonium mixture and a phosphate concentration of 1 g/L. The use of this culture medium makes it possible to obtain an average specific production of surfactin of 83.3 mg/g compared to 42 mg/g for the control medium, i.e., an increase of approximately 2 times. The use of mannitol in combination with the urea + ammonium mixture gives also interesting results. On the other hand, the lowest productivity is observed for media using glycerol.

Based on these results, it was decided to produce the FS mixture by *B. subtilis* ATCC21322 using a medium composition as follow: glycerol, 40 g/L; glutamic acid, 5 g/L; KH_2_PO_4_ 1 g/L, and others Landy medium components. Results of the production carried out in 5 L Erlenmeyer flask were 163 ± 23 mg/L for fengycin and 145 ± 31 mg/L for surfactin. These results were calculated on 6 different productions. Productions were pooled to obtain solution with a ratio of FS of 55–45 %. This result was very close to the expected one (i.e., 50–50%).

### 3.3. Protection of Apple Tree against Scab

The different lipopeptide mixtures were then tested in orchards during two trial seasons in 2018 and 2019 and compared to different modalities: a water control, adjuvant 1 and 2 modalities added during the spraying of the lipopeptide mixtures, a copper/sulphur modality and SERENADE^®^ product. Analysis of variance was performed on the leaf and fruit data collected from the various scab surveys during the primary disease cycle. The results presented are based on the findings of the leaf analysis at the last survey, i.e., at the end of the primary scab infections.

#### 3.3.1. Results from 2018 Trial

The 2018 results are presented in Figure 6. The analysis of variance carried out allows us to conclude with a high probability (*p* value = 0.02) that there are significant differences between the modalities studied. The different modalities are divided into two distinct statistical groups. The first group (A) alone includes the M1 water control modality with the highest rates of contamination on leaves by scab. The second group (B) is composed of the modalities with the lowest rates of contamination by scab, namely: the M7 SERENADE^®^ modality, the M4 lipopeptides 2 modality composed of the mixture of MS, the M4 lipopeptides 1 modality composed of the mixture of FS and finally the M6 modality of biological reference.

Statistically, under the conditions of the 2018 study, the two modalities composed of a mixture of lipopeptides (M4 and M5) had a behaviour similar to the modality M6 of biological reference and M7 of Biocontrol (SERENADE^®^). The effectiveness of these two modalities (M4 and M5), although very similar, was not superior, at the end of the primary contaminations, to that of the reference modality (M6 or M7). During the primary post-flowering contaminations under the conditions of this trial, the efficacy of the lipopeptide mixtures was occasionally higher than the trial reference, ranging from 0 to 16% for the MS mixture and from 0 to 58% for the FS mixture. In contrast, it consistently ranged from 26 to 73% for the MS mixture and from 50 to 69% for the FS mixture, compared to the water control (data not shown).

#### 3.3.2. Results from 2019 Trial

The 2019 results are presented in Figure 7. The analysis of variance carried out allows us to conclude with a very high probability (*p* value < 0.001) that there are very highly significant differences between the modalities studied. The different modalities are divided into two distinct statistical groups. The first (A) includes the M1 water control, M2 lipopeptide 1 adjuvant control FS and the M4 lipopeptide 1 modality composed of FS. This group had the highest rates of leaf scab contamination in the test. In a second group (B), the modalities M6 biological reference, M7 fertilizer and the modality M5 lipopeptides 2 composed of MS are combined. This group has the lowest levels of scab contamination. In the meantime, only the M3 adjuvant lipopeptide control MS belongs statistically to both groups. It is possible to observe a more effective behaviour, under the experimental conditions of the year, of the lipopeptide mixture composed of MS. This last modality showed a contamination rate comparable to the reference modality M6 of the trial.

The FS mixture, used at a lower concentration than the MS mixture, showed a slow and regular loss of efficacy during primary infections which certainly explains its low efficacy results in this trial (Appendix A). The efficacy of each of these two modalities, although dissimilar, was not higher at the end of the primary infections than the reference modality.

In contrast, the MS mixture showed a 35% higher efficacy than the reference at the very beginning of the disease cycle (Appendix A). Compared to the water control, the MS mixture consistently showed a relatively constant efficacy ranging from 72 to 60%. The FS mixture, however, showed a decrease in efficacy over time from 68 to 22% at the end of the primary infections.

### 3.4. Study of the Persistence of Molecules after Spraying on Fruit Trees

QuEChERS method was first validated by adding powder of lipopeptides (surfactin, fengycin or mycosubtilin) to a crushed apple preparation before applying the QuEChERS method solid phase extraction in order to ensure that the three lipopeptides were well detected (data not shown). The calibration curves obtained for the three lipopeptides allowed to quantify them on skin and flesh of the different apples between 0.004 to 0.066 mg of lipopeptides by kg of apple. Quantification of the three families of lipopeptides was performed by RP-UPLC-MS to study the persistence of lipopeptides 45 days after the spraying on fruit trees. Control samples (untreated), samples treated with the MS mixture, samples treated with the FS mixture were analyzed. Quantification of lipopeptides is quite complex because of the numerous forms (the different protonated isoforms as well as the sodium or potassium adducts) detected by mass spectrometry. All the forms were analyzed in this experiment, but quantification was only performed on the most intense ions. Results are presented in Table 4.

The results from the QuEChERS method, presented in Table 4, reveal the presence of only fengycin and surfactin on the skin of the apple; mycosubtilin is not present either on the fruit skin or in the flesh. The presence of fengycin in the skin and the flesh is below the acceptable threshold of 0.01 mg/kg plant material. However, surfactin was quantified at concentrations above the recommended limit for a treatment product in skin and flesh. Note that, for some replicates, the lipopeptide was not detected (data not shown). The differences obtained between the replicates come from the complexity of the apple matrix.

## 4. Discussion

### 4.1. In Vitro Antifungal Activities of the Lipopeptides Mixtures

The objective of this study was to find the best mixture of lipopeptides produced by a natural strain of *B. subtilis* in order to apply it in organic orchards to control *V. inaequalis*. Fengycin, mycosubtilin and surfactin show different antifungal activity against *V. inaequalis* (F = 0.05 mg/L (0.03–0.07); M = 1.38 mg/L (1.15–1.68); S of the mixture FS = 6.38 mg/L (5.30–7.67) or S of the mixture MS = 4.79 mg/L (3.81–6.01)). The results agree with [23] where similar activities were obtained for each lipopeptide (F = 0.03 mg/L; M = 2.15 mg/L; S = 5.98 mg/L). In addition, the antifungal activity of surfactin against *V. inaequalis* is confirmed as described in the literature [25,26]. FS and MS mixtures with a majority of fengycin or mycosubtilin show interesting but equally effective antifungal activity as 100% fengycin or 100% mycosubtilin. It has already been demonstrated that MS and FS mixtures have antifungal activities on different phytopathogens with synergetic effect [27,28,39,56]. The antifungal activity of different microorganisms against apple scab has already been studied at a lab scale or in field. Indeed, in vitro tests of isolates of *Pseudomonas*, *Trichoderma* and *Bacillus* show an inhibition of *V. inaequalis* [57,58]. Their antagonistic activity towards *V. inaequalis* is variable (11 to 58% inhibition for *Pseudomonas,* 100% for *Trichoderma* and 33 to 41% for *Bacillus*). Field studies have been conducted using *Trichoderma* spp. and *Bacillus* spp. showed encouraging results in decreasing the incidence and severity on fruits and foliage [57]. Another microorganism, *Cladosporium cladosporioides*, showed an antagonistic effect, also by decreasing the incidence of apple scab on fruit and foliage [59,60]. All these works suggest the presence of secondary metabolites to explain the microorganism activity. The results of our study on purified lipopeptides show, without any doubt, the involvement of these molecules in the protection against apple scab. It also confirms the results we obtained in vitro [23]. The results on the ratio of the molecules when using a mixture of lipopeptides show the importance of being able to control and orient the metabolism of the strain for the production of the best mixture and thus increase the effectiveness of the biocontrol preparation. This type of lipopeptide mixture is found in many *Bacillus* biocontrol products [61].

### 4.2. Production of the Fengycin/Surfactin Mixture and Its Optimization by Design of Experiment

The results of the experimental designs show that fengycin production by the *B. subtilis* strain ATCC 21332 is significantly impacted by the carbon source. Glycerol and to a lesser extent mannitol allow the overproduction of fengycin compared to the use of glucose. These results are in agreement with a previous study in 2016, which shows that mannitol allows for the best productivity of fengycin compared to 11 other carbon sources using the strain BBG 21 [45]. They also go in the same direction as those presented the same year in another work, where the specific production of fengycin of the *B. amyloliquefaciens* 0G strain is better in a medium containing the glycerol–glutamic acid combination in comparison with a glucose–glutamic acid [47]. The impact of low K_L_a which favours fengycin production agrees the previously published results with BBG21 [43] and ATCC 21332 strains [44]. It is interesting to observe that the optimization results obtained at the mL scale were confirmed with only little deviation at the liter scale. These results are a reminder of the value of using DoE for medium optimization but also of the robustness of the high throughput system used here (i.e., Biolector^®^ supplied by m2p-labs GmbH, Germany [62]). In terms of culture medium optimization, future studies are needed to optimize the concentrations of the constituents identified in our study. The use of Response Surface Methodology will be considered, as we had previously done for the optimization of surfactin with another strain of *Bacillus* [49]. The use of glycerol as a carbon source can be part of a circular economy approach that allows this carbon source to be valorized. Recent studies have highlighted this aspect for the production of *Bacillus* lipopeptides [47,63,64]. Particular attention should however be paid to the grade of glycerol used, as this can have a direct impact on the profile of the fengycin isoforms produced and therefore in-fine the activity of the antifungal preparation [65].

### 4.3. Protection against Apple Scab

During the two different campaigns of this study, it was possible to observe a good behaviour of lipopeptide mixtures in the protection against apple scab. Specially, the lipopeptide mixture composed of mycosubtilin (80%) and surfactin (20%) at the dose of 500 mg/L showed a significant effectiveness, close to the reference (Cu/S), throughout the disease cycle. In contrast, the lipopeptide mixture of fengycin (55%) and surfactin (45%) at the lower dose of 250 mg/L appears to provide protection only under low disease pressure, as we observed during the 2018 trial, with the former diminishing at higher disease pressure. These results call in the first place for an increase in the dose of this preparation when treating orchards. This preparation is half as concentrated in lipopeptides as the mycosubtilin/surfactin preparation. In these results, it should be noted that the formulation adjuvants also have a slight effect against *V. inaequalis*, sharing the statistical groups with the modalities of mixture of control and lipopeptides. This very slight indirect effect is noticeable due to the relatively short duration of the 2018 trial. With the lengthening of the duration of the trial, this effect tends to disappear and is no longer perceptible like the tests conducted in 2019. The addition of the “adjuvant control” modalities allows us to ensure the direct protective effect of the substances tested. The antifungal action of this type of terpenic adjuvant has already been described in the literature, especially in combination with essential oils. Another important point is the formulation of these complex amphiphilic molecules which must remain on the leaves and not be washed away by the first rain. To solve this problem, the lipopeptides formulation must be further developed, either by optimizing the concentrations of the adjuvants we have used in this work, or by using other more effective sticker molecules. It is also necessary to study their integration into a disease control program. As shown by these first trials, protection against apple scab using lipopeptides alone still seems optimizable to date, with efficiencies often lower than the reference (i.e., Cu/S). Finally, the use of a mixture combining a very antifungal molecule such as mycosubtilin or fengycin with surfactin in orchard treatments can have many major advantages. The first is to benefit from the synergistic effect against the phytopathogen. The second is that surfactin could have a stimulating effect on plant defence, as demonstrated in other plants [31]. This has not been reported in apple against *V. inaequalis* but it is a promising way of study for future studies on this pathosystem. The third is to provide an additional insecticidal effect against one of the main pests of apple trees. Indeed, in a recent study, authors have shown that surfactin displayed an aphicidal activity against *Dysaphis plantaginea* or *Aphis fabae* even at low concentration [66,67].

### 4.4. Persistence of Lipopeptides on Apple Fruits

The persistence of the different lipopeptide mixtures was evaluated 15 days after fruit harvest. The results from the QuEChERS method did not reveal the presence of mycosubtilin either on the fruit skin nor in the flesh. However, fengycin and surfactin were found. The quantification of lipopeptides from apple extract had already been carried out in work on the use of a *Bacillus subtilis* GA1 strain for the protection of fruit against *B. cinerea*, fengycin being the main lipopeptide found [68]. In our results, fengycin was only found in the flesh and skin at a concentration well below the detection limit of the method. Surfactin was recovered on the apple’s skin and flesh at a concentration above the recommended limit for a treatment product. The high surfactant activity of surfactin seems to allow it to penetrate well into the fruit, which does not seem to be the case for the other two families of lipopeptides. This result is to be highlighted with regard to the known hemolytic power of surfactin but also to its known cytotoxicity against cells of the digestive system like Caco-2. A recent study shows that the IC_50_ is around 200 mg/L on these intestinal cells [69]. The results of remanence show values well below the cytotoxic thresholds.

## 5. Conclusions

Lipopeptides from *B. subtilis* have shown, for the first time, in vivo antifungal efficiency on *V. inaequalis* in orchard trials. These results are particularly impressive with mycosubtilin/surfactin mixture, allowing a reduction of the disease incidence by about 60%, and to a lesser extent with fengycin/surfactin mixture. It is also interesting to observe that surfactin has an antifungal effect against *V. inaequalis* even if it is 4 to 100 times less powerful than with the other two families of lipopeptides. Nevertheless, the combination of fengycin or mycosubtilin with surfactin may be of interest as it was demonstrated in vivo in this study for its synergistic effect, its potential as a resistance inducer but also its aphicidal activity.

## Figures and Tables

**Figure 1 microorganisms-10-01810-f001:**
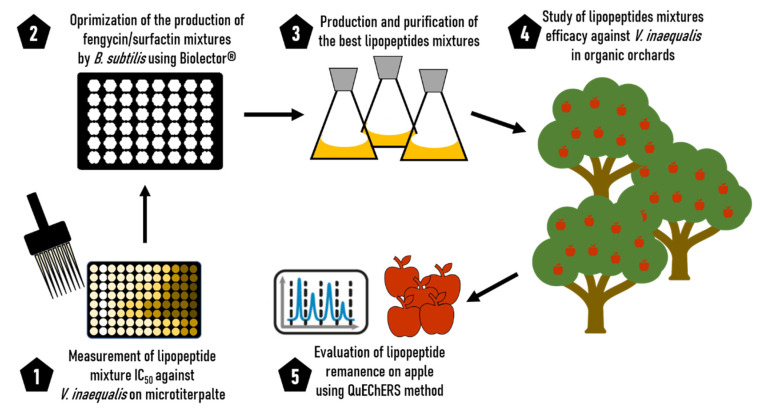
Schematic representation of scientific strategy developed in this work. Step 1: in vitro screening of the best lipopeptide mixture against *V. inaequalis*; Step 2: Optimization of the lipopeptide mixture production; Step 3: Production and purification of the different mixtures of lipopeptides in shacked flask; Step 4: Study of lipopeptide mixture efficiency against *V. inaequalis* in organic orchards; Step 5: Lipopeptide remanence evaluation using QuEChERS method.

**Figure 4 microorganisms-10-01810-f004:**
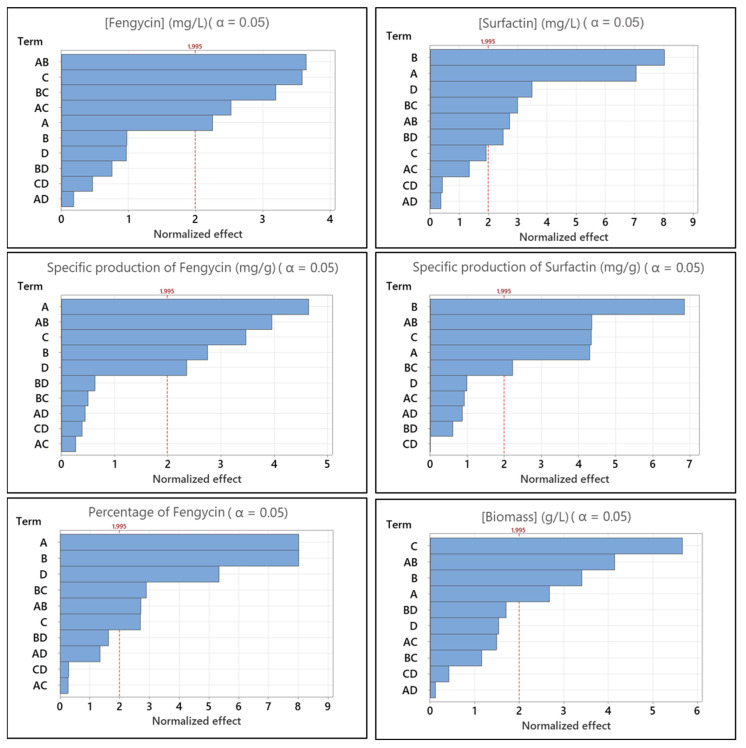
Pareto diagram of normalized effect for fengycin concentration, surfactin concentration, specific production of fengycin and specific production of surfactin, biomass production and percentage of fengycin produced in relation to total lipopeptides produced. A: carbon source; B: Nitrogen source; C: Phosphate concentration; D: K_L_a.

**Figure 5 microorganisms-10-01810-f005:**
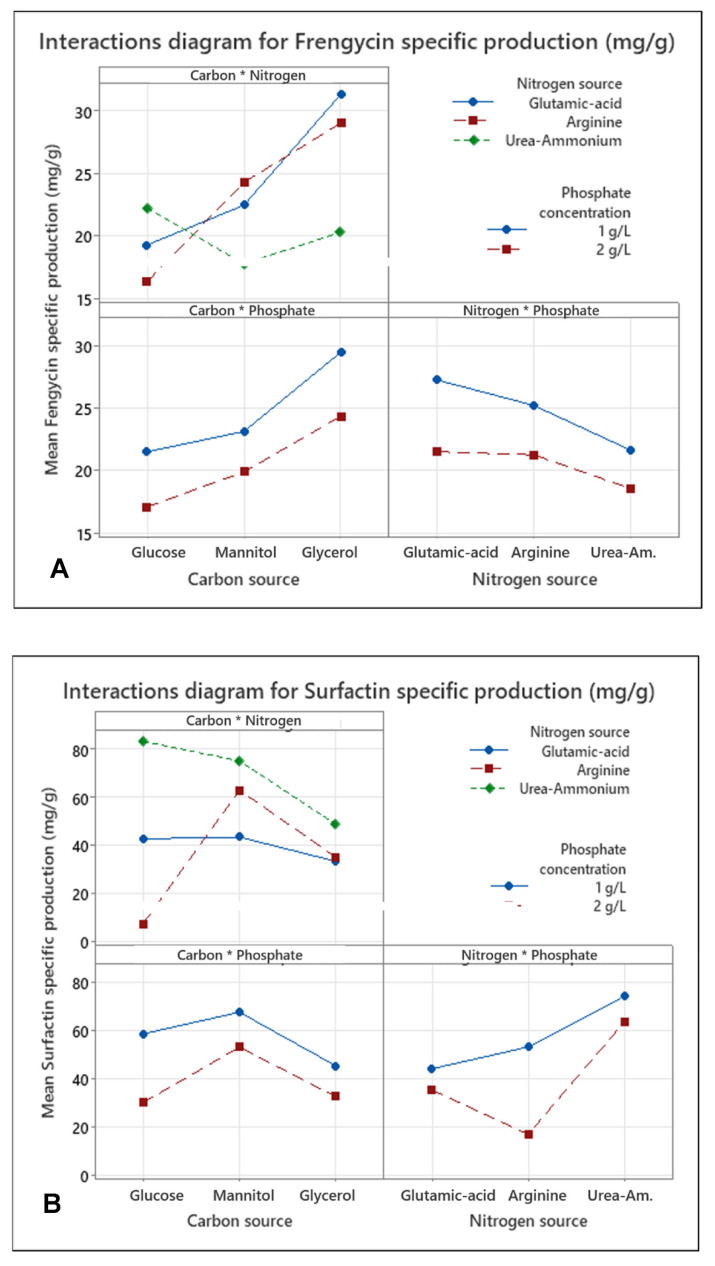
Interaction diagrams for (**A**) fengcyin specific production, (**B**) surfactin specific production and for (**C**) Percentage of fengycin produced related to total lipopeptides. * indicates the interaction between the factors.

**Figure 6 microorganisms-10-01810-f006:**
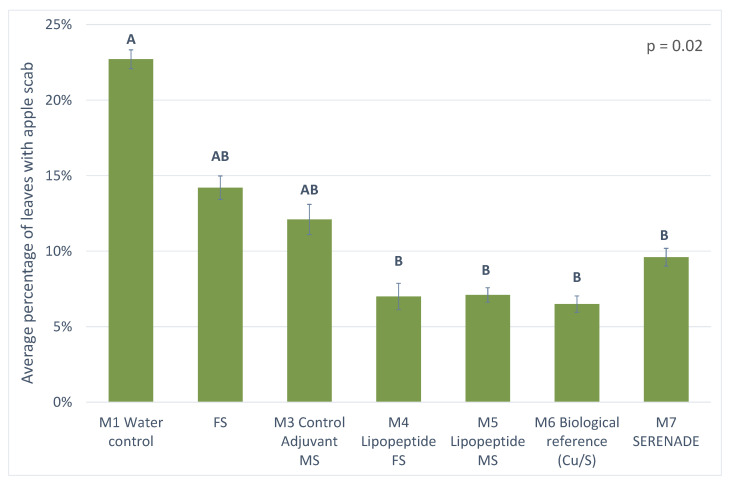
Graphical representation of the levels of leaf scab contamination in the different modalities following the 2018 trials at the end of primary contamination. The different statistical groups are represented by the letters A and B. A rating of AB indicates that this modality belongs statistically to groups A and B.

**Figure 7 microorganisms-10-01810-f007:**
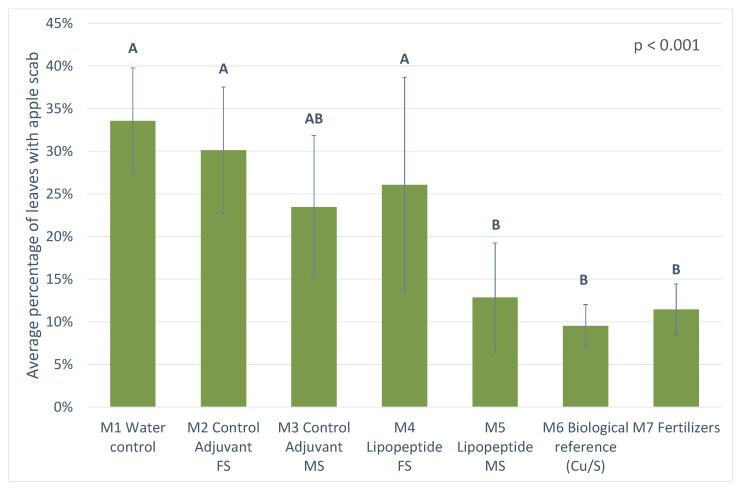
Graphical representation of the levels of leaf scab contamination in the different modalities following the 2019 trials at the end of primary contamination. The different statistical groups are represented by the letters A and B. A rating of AB indicates that this modality belongs statistically to groups A and B.

**Table 1 microorganisms-10-01810-t001:** Lipopeptides used in this study, their ratios and purity.

Lipopeptide(s)	Ratios (%)	Code	Purity (%)
Fengycin	100–0	F100	98
Fengycin–Surfactin	80–20	F80–S20	98–90
	60–40	F60–S40	
	50–50	F50–S50	
	40–60	F40–S60	
	20–80	F20–S80	
Surfactin	0–100	S100	90
Mycosubtilin	100–0	M100	75
Mycosubtilin–Surfactin	80–20	M80–S20	75–90
	60–40	M60–S40	
	50–50	M50–S50	
	40–60	M40–S60	
	20–80	M20–S80	
Surfactin	0–100	S100	90

**Table 2 microorganisms-10-01810-t002:** Summary of the modalities studied against apple scab in 2018.

Modalities	Pre-Flowering			Post-Flowering		
	Substances	Dose	Applications	Substances	Dose	Applications
M1 Water control	Water		T1 to T5	Water		T6 to T11
M2 Control Adjuvant Lipopeptides 1 (Fengycin/Surfactin)	Copper	1.5 kg/ha	T1 to T2	Adjuvant	2 L/ha	
M3 Control Adjuvant Lipopeptides 2 (Mycosubtilin/Surfactin)				Adjuvant		
M4 Lipopeptides 1 (Fengycin/Surfactin)	Sulphur	7.5 kg/ha	T3 to T4	55–45%	250 mg/L	
M5 Lipopeptides 2 (Mycosubtilin/Surfactin)				80–20%	500 mg/L	
M6 Biological reference (Cu/S)	Sulphur + Copper	5 kg/ha + 1 kg/ha	T5	Sulphur + Copper	5 kg/ha + 1 kg/ha	
M7 SERENADE				*B. subtilis* str. QST 713	2 kg/ha	

**Table 3 microorganisms-10-01810-t003:** Summary of the modalities studied against apple scab in 2019.

Modalities	Substances	Dose	Applications
M1 Water control	Water	-	T1 to T12
M2 Control Adjuvant Lipopeptides 1 (Fengycin/Surfactin)	Adjuvant	2 L/ha	T1 to T12
M3 Control Adjuvant Lipopeptides 2 (Mycosubtilin/Surfactin)	Adjuvant	-	T1 to T12
M4 Lipopeptides 1 (Fengycin/Surfactin)	55–45%	250 mg/L	T1 to T12
M5 Lipopeptides 2 (Mycosubtilin/Surfactin)	80–20%	500 mg/L	T1 to T12
M6 Biological reference (Cu/S)	Copper	1.5 kg/ha	T1
	Sulphur	7.5 kg/ha	T2 and T4
	Sulphur + Copper	5 kg/ha + 1 kg/ha	T5 to T12
M7 Fertilizers	Copper	1.5 kg/ha	T1
	Fertilizer 1	2% to 4%	T2 to T4
	Fertilizer 2	8%	T5 to T12

**Table 4 microorganisms-10-01810-t004:** Quantification of fengycin, surfactin and mycosubtilin following extraction by the QuEChERS method in the different samples by RP-UPLC-MS ^1^.

Modalities		Fengycin	Surfactin	Mycosubtilin
Control	Skin			
	Flesh			
Fengycin–Surfactin	Skin			
	Flesh			
Mycosubtilin–Surfactin	Skin			
	Flesh			

^1^ Cells in black correspond to a concentration of corresponding lipopeptide superior to the acceptable threshold between 0.01 mg/kg and 0.03 mg/kg of apple, cells in dark grey to a concentration between 0.004 and 0.01 mg/kg of apple, cells in light grey to a concentration inferior to 0.004 mg/kg of apple and cells in white means that lipopeptide is not detected in these experimental conditions.

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
