# Peer review of "Assessment of Lipopeptide Mixtures Produced by Bacillus subtilis as Biocontrol Products against Apple Scab (Venturia inaequalis)"

_microorganisms, 2022, doi:10.3390/microorganisms10091810_

Round 1

Reviewer 1 Report

This is a comprehensive report, from producing the lipopetides and testing their activities in vitro, to evaluating their disease control and confirming their persistence in vivo. The objectives of the research are relatively clear and the presentation of the results and interpretation of the data are generally appropriate although the experimental design, sampling /data collection, particularly, procedures used for statistic analyses could be better described.

The following are more specific comments / suggestions:

Lines 20 and 22: spell out B. subtilis and V. inaequalis; also other names of organisms when they are appeared for the first time in the MS.

Table. 1: the purity of mycosubtilin was only 75%. What was the other 25% possibly? Were all the active ingredients adjusted according to their purity? Why?  The same question for all the experiments involving the lipopeptides.

Line 164-165 / Line 342-349: the procedures applied for the statistical analyses are indeed more important than the software used, should be indicated / described.   Such descriptions are also missed for other experiments, e.g. Fig 2,Fig 3, … … A paragraph describing statistical analyses for all the experiments is necessary in the M&M section.

Line 224: “a Fisher block design with four replicates and no control” – water control?

Fig 6 and Fig7: “Average percentage of leaves with apple scab” -  how the disease was evaluated, any consideration of disease severity?  Such evaluations are significantly important for the conclusions of disease control efficiency, thus should be described and discussed clearly.

Author Response

Thank you for your comments and suggestions on our work, you will find attached a document with our answers point by point

Sincerely yours

Reviewer 2 Report

Comments

This manuscript titled ”Assessment of lipopeptide mixtures produced by Bacillus subtilisas biocontrol products against apple scab (Venturia inaequalis)investigated the effect ofthe three families of lipopeptides on V. inaequalisin vitroand in vivo. This work is heavy and interesting, but not be systematic. In addition, some errors exist in the present version. Thus, this manuscript should be modified before publishing in this journal.

Specific points are issued as follows.

1/Ln 24,what means DoE here?

2/The modifications should not be shown in this manuscript, such as Ln128/192/354.

3/ Ln 137-138, are the powdered lipopeptides purified by authors? Not clear here. 

4/Ln139, “FS and MS” should be cleared when they appear in the first time. 

5/ Ln 199-200, please check the formats of chemicals.

6/ Ln 203-204, the common purification process is not described clearly, and the extraction process should be checked and supplemented with present references. 

7/ why use the mixture of lipopeptides such as M3, M4, and M5 shown in table 2?

8/ Ln 292, the substances of m/z 753.44,1072.69, and 1107.58 are these three lipopeptides? Check it, and the chromatogram should be supplemented.

9/ Figure 6, this figure is not presented clearly here, and significant level P=0.02? 

10/Ln 503-504, how to support this point?

11/Ln 574-576, how to find a solution to this problem in authors’ experiments?

12/ the Figure S3, the letters in this figure are confusing for reading. The date should be checked and the letters should be noted clearly.

13/For the part References, theLatin name of each species should be spelled in italic, and the format of each reference should be unified and checked carefully.

Author Response

(The authors gave the same response as above.)

Reviewer 3 Report

The experiments are well designed and the structure of the manuscript is well organized. However, some corrections are necessary. For example, in line 77, should be "activities"; line 93 should use the past tense; in line 386, should be "a good" instead of "an good"...It would be better if the manuscript can be revised by an English speaker.

Author Response

(The authors gave the same response as above.)

Round 2

Reviewer 2 Report

Authors have modified this manuscript, no more comments.